# Effect of Novel Gastro-Duodenal Flow Restrictor on Relative Weight Loss and Glucose Levels in a Porcine Model: A Pilot Randomized Study

**DOI:** 10.3390/nu14132563

**Published:** 2022-06-21

**Authors:** Gunn Huh, Jinhee Kwon, So Hee Kim, Ha Jong Lim, Se Hee Min, Do Hyun Park

**Affiliations:** 1Division of Gastroenterology, Department of Internal Medicine, University of Ulsan College of Medicine, Asan Medical Center, Seoul 05505, Korea; drthevision@gmail.com (G.H.); jinheekwon.bme@gmail.com (J.K.); lucia468286@gmail.com (S.H.K.); jongjong0308@gmail.com (H.J.L.); 2Department of Medical Science, Asan Medical Institute of Convergence Science and Technology, University of Ulsan College of Medicine, Asan Medical Center, Seoul 05505, Korea; 3Division of Endocrinology and Metabolism, Department of Internal Medicine, University of Ulsan College of Medicine, Asan Medical Center, Seoul 05505, Korea; shminmd@gmail.com; 4Digestive Diseases Research Center, Department of Internal Medicine, University of Ulsan College of Medicine, Asan Medical Center, Seoul 05505, Korea

**Keywords:** obesity, metabolic disease, endoscopic bariatric and metabolic therapies, gastro-duodenal flow restrictor, stent, relative weight loss

## Abstract

Endoscopic bariatric and metabolic therapies are promising for obesity. We developed a novel gastro-duodenal flow restrictor (G-DFR) device for relative weight loss and lowering of glucose level and evaluated its safety and efficacy in a porcine model. The G-DFR comprised self-expandable gastro-duodenal partially covered polytetrafluoroethylene (PTFE) metal stent distally attached to a PTFE skirt. Eleven juvenile pigs were randomized into the evaluation of migration (*n* = 3), mid-term efficacy (*n* = 5), and control (*n* = 3) groups. Five pigs showed G-DFR migration at 2, 4, 7, and 10 weeks after placement in the migration and mid-term efficacy group. Compared to the control group, the mid-term efficacy group showed up to 55.4% relative weight loss in 12 weeks. Compared to the case group, the control group showed higher mean ghrelin hormone level from 6 to 12 weeks. Glucose level was significantly lower in the efficacy group than in the control group after 6 weeks. Serum alanine transferase levels and histological collagen deposition were lower in the liver of the case group than in the control group. Although it did not demonstrate consistent performance with respect to migration, a well-positioned G-DFR in the pyloroduodenal portion may lead to relative weight loss, lowering of glucose levels, and improved hepatic parameters.

## 1. Introduction

The prevalence of obesity has increased rapidly over the last 50 years, and it has now become a major global health issue [1]. Therapeutic options for obesity include lifestyle modifications, medications, and bariatric surgery. Anti-obesity drugs have limited efficacy and are often accompanied by gastrointestinal or neuro-psychiatric adverse events [2]. Bariatric surgery, such as sleeve gastrectomy and Roux-en-Y gastric bypass, is the most effective treatment for severe obesity; however, only a few patients are subjected to these procedures because of their invasiveness and irreversibility [3].

Endoscopic bariatric and metabolic therapies (EBMTs) have emerged as promising alternative options for the treatment of obesity and metabolic disorders, such as type 2 diabetes and nonalcoholic fatty liver disease [4,5,6]. Most procedures such as intragastric balloons, endoscopic sleeve gastroplasty, primary obesity surgery endoluminal, aspiration therapy, and transpyloric shuttle predominantly target the stomach, while other procedures such as endoscopic duodenal–jejunal bypass liner and duodenal mucosal resurfacing focus on the duodenum [7,8,9]. The principle of gastric EBMTs is based on the stimulation of gastric mechanical and chemical receptors, delay of gastric emptying, and modulation of gastric orexigenic hormones [7]. Duodenal EBMTs have garnered interest as promising treatment options for metabolic disorders, such as diabetes and nonalcoholic fatty liver disease, as well as obesity, because the proximal small bowel is known to play a crucial role in glucose homeostasis [10]. 

We developed a novel endoscopic device, gastro-duodenal flow restrictor (G-DFR) to reduce weight gain rate and lower glucose levels. The theoretical advantage of this novel device is that it targets both the stomach and the duodenum, thereby maximizing the effect of weight loss and metabolic improvements. The hypothetical mechanisms underlying the mode of action of this device include delayed gastric emptying and bypass of the proximal duodenum. The aim of this study was to evaluate the feasibility, safety, and efficacy of G-DFR in a porcine model. 

## 2. Materials and Methods

### 2.1. Animal Study 

This study was approved by the Institutional Animal Care and Use Committee of our institution (2021-12-049). The study conformed to the guidelines of the University of Ulsan for humane handling of laboratory animals. 

A total of 11 Yorkshire female pigs weighing 31.0–36.5 kg (median, 33.8 kg) (International experimental animal, Pocheon, Korea) were randomly divided into three groups. Three pigs were evaluated for migration of G-DFR on days 14, 28, and 42 (evaluation of migration group). Five pigs underwent stent placement (evaluation of mid-term efficacy group). The placement was observed for 90 days in the mid-term efficacy group. The remaining three pigs were assigned to the control group (Figure 1).

All pigs were fed a diet comprising regular chow once a day. The intake of food was gradually increased till each pig was fed 3 kg of regular chow for 70 days. All pigs were maintained at 22 ± 2 °C.

### 2.2. Novel G-DFR Delivery System

This novel G-DFR device comprised a partially covered metal stent of polytetrafluoroethylene (PTFE) membrane, an uncovered proximal flap of 50 mm and a distal flap of 30 mm, in its outer diameter for anchoring the device to the pyloric and duodenal sections, respectively. It also comprised a stent with a diameter of 15 mm, a length of 60 mm, and an embedded PTFE membrane with a 60 mm distal PTFE skirt. A higher delayed gastric emptying time has been reported with a gastroduodenal stent of 18 mm diameter compared with a stent of 20 mm diameter [11]. Therefore, a gastroduodenal stent with 15 mm lumen diameter may further restrict flow of gastric contents and delay gastric emptying. We attached a 60 mm PTFE skirt in the distal part of the stent as a flow restrictor. This PTFE skirt may further restrict gastro-duodenal flow due to friction (Figure 2, left). In a previous study on in vitro evaluation with a 3D human peristaltic gastroduodenal simulator [12], the gastric emptying volume of G-DFR with 15 mm lumen diameter and a 60 mm PTFE skirt at the distal end of G-DFR was the least, suggesting enhanced gastric retention compared with intragastric balloon, duodenal stents of 18 and 20 mm lumen diameter, and G-DFR with a 30 mm PTFE skirt. Therefore, we chose the 15 mm lumen diameter of G-DFR with 60 mm PTFE skirt for further delaying gastric emptying in this pilot animal study.

A retrievable lasso was placed on the proximal flap of G-DFR. When a lasso was grasped and pulled via grasping forceps, G-DFR could be flattened, which facilitated to remove the device in uncovered portions. This G-DFR with the diameter of 10.5 Fr in the stent introducer was not commercially available (Figure 2, right).

### 2.3. Procedure of G-DFR Placement and Removal

After 24 h of fasting, the pigs were intramuscularly administered 5 mg/kg of zoletil and 1.5 mg/kg of xylazine under the supervision of a veterinarian. An endotracheal tube was placed, and anesthesia was administered via inhalation [2.5–3% isoflurane (Ifran^®^; Hana Pharm. Co., Seoul, Korea) with oxygen (2 L/min) at 1:1]. All procedures were performed in the left lateral decubitus position. An overtube was placed through the mouth into the esophagus. A linear echoendoscope (EG-580UT with 3.8 mm working channel diameter; Fujifilm Medical, Tokyo, Japan) or a duodenoscope (TJF-260V with 4.2 mm working channel diameter; Olympus Inc., Tokyo, Japan) was introduced through the overtube into the stomach. A 0.025-inch guidewire (Visiglide 2, Olympus Inc.) and a catheter (Glo-tip, GT-1-T, Cook medical, Bloomington, IN, USA) were passed through the working channel of the endoscope into the stomach. The catheter was removed while retaining the guidewire, and the G-DFR delivery system was transferred over the guidewire into the stomach. The G-DFR was situated at the junction of the pylorus and the upper duodenum. After removing the G-DFR delivery system, the balloon (CRE balloon (Boston Scientific, Cork, Ireland) or Ren balloon (Kaneka, Osaka, Japan)) dilation was performed to facilitate G-DFR placement via a guidewire at the discretion of an endoscopist. Two or four hemoclips (Hilzo clip, BCM, Goyang, Korea; Optimos clip, Taewoong medical, Gimpo, Korea; or Sure clip, Micro-Tech, Nanjing, China) were used to anchor the proximal flaps of G-DFR. [12] In due time, G-DFR was endoscopically removed by rat-tooth forceps (FG-26-C1, Olympus Inc.) that grasped the lasso in the proximal flap.

### 2.4. Follow-Up

After G-DFR placement, the pigs were fed after recovery from the anesthesia. The case pigs were followed up based on 14 days (evaluation of migration *n* = 1), 28 days (evaluation of migration *n* = 1), 42 days (evaluation of migration *n* = 1), and 90 days (evaluation of efficacy *n* = 5) observation after placement of G-DFR. All pigs survived until fulfillment of follow-up periods in each group except the case with migration. (Figure 1) The interpreted data representing the case trend of weight and biochemical value were only selected based on the monitoring time point before possible migration was noted.

Weight and behavioral changes and the levels of fasting blood glucose, ghrelin, insulin, alanine aminotransferase (ALT), and cholesterol were monitored after two days of G-DFR placement every two weeks until end of planned follow-up. Total (acyl and des-acyl) ghrelin (AFG bioscience, Northbrook, IL, USA) and insulin (AFG bioscience) levels were measured before the morning meal at 9 a.m. [8].

All pigs were euthanized via the administration of an overdose of KCl (Sigma, St. Louis, MO, USA) after planned follow-up periods or migration of G-DFR. Surgical exploration of the pyloric portion of the stomach, duodenum, and liver was followed by gross examination to evaluate the degree of the formation of granulation tissue. Masson’s trichrome (MT) staining was performed to evaluate interstitial collagen volume in the liver during fibrosis evaluation and to determine possible pyloric portion. Duodenal injury after G-DFR placement was examined by microscopic evaluation with hematoxylin and eosin staining.

The paraffinized sections (thickness of 3 μm) were processed with Masson’s Trichrome (MT) staining to examine interstitial collagen deposition and the corresponding occupied regions. The contrast was observed with the red-stained parenchyma and blue-stained fibrous tissue, indicating collagenous region. The sections were digitalized using a slide scanner (VS200; Olympus). Images were analyzed using image analysis software (VS20S-DESK v3.2; Olympus) to quantify the collagen deposits, and their average values were assessed. We calculated the percentage of the area occupied by collagen (blue) in regions with a predominance of fibers cut in the longitudinal plane (red and blue).

### 2.5. Primary and Secondary Outcomes

The primary endpoint was the relative weight loss in the case group compared to that in the control group observed during 90 days of follow-up. Relative weight loss was considered as the reduction in the proportional change in body weight, which was calculated using the following formula: (Body weight change in control [%] − Body weight change in case [%]) [13]. The proportional change in body weight was defined according to the following formula: (gained weight − baseline weight)/baseline weight × 100. Secondary outcomes were relative weight loss without migration of G-DFR over 56 days (mid-term follow-up), levels of serum glucose, ghrelin, and insulin, chemistry, liver fibrosis, adverse events, and safety profiles. 

### 2.6. Statistical Analysis

Since this was a pilot randomized study, the sample size was not calculated. Continuous variables are expressed as mean values with standard deviation or as median values with an interquartile range (IQR). A sample *t*-test was used to evaluate the relative weight loss in the case group when compared to that in the control group. For all analyses, a one-sided *p*-value of < 0.05 was considered to indicate a statistically significant difference. All statistical analyses were performed using GraphPad Prism 9.0.0 software (GraphPad Software Inc., San Diego, CA, USA).

## 3. Results

### 3.1. Technical and Safety Profile of G-DFR

Endoscopic placement of G-DFR was technically successful in all 11 pigs. When the flow of gastroduodenal fluid in G-DFR was evaluated, the injected fluid slowly ran through distal PTFE skirt of G-DFR after placement of G-DFR (Appendix A). No adverse event was observed in all 11 pigs during endoscopic procedure. The procedure duration was 16 to 31 min (median 25.8 min (IQR 24–30)).

In the evaluation of migration group, one of the three pigs showed distal migration of G-DFR. The remaining two pigs showed no migration until the end of the follow-up periods (28 and 42 days). In the evaluation of mid-term efficacy group, three pigs with endoscopic placement of G-DFR without balloon dilation showed distal migration of G-DFR during the follow-up periods. Two pigs with endoscopic placement of G-DFR with balloon dilation showed no migration after 56 days. However, only one pig did not demonstrate any migration until the end of the follow-up period (Figure 3). Overall migration rate was 33% in the evaluation of migration group (*n* = 1/3) and 80% in the evaluation of mid-term efficacy group (*n* = 4/5). One was a distal migration without becoming lodged in the bowel. Remaining four was a partial migration in the distal duodenum, and the migrated G-DFR in these cases was endoscopically removed. No difference was observed in G-DFR migration with respect to the number of hemoclips (two or four) required for anchoring the proximal flaps of G-DFR. Endoscopic removal of G-DFR was successfully performed at 28 days, 42 days, and 90 days. Because of the difficult interpretation, cases involving migration were excluded, and cases without the migration of G-DFR (*n* = 1 in 90 days as a primary outcome; *n* = 1 in 90 days and *n* = 1 in 56 days as a mid-term efficacy outcome shown in Appendix A) were only evaluated for the interpretation of efficacy of G-DFR. No acute adverse event was observed during the endoscopic removal of G-DFR (Figure 3, upper). During necropsy in the G-DFR group, no obvious duodenal perforation and peritoneal inflammation were observed (Figure 3, middle). Histological analysis showed granulation tissue formation and shallow ulceration but no perforation in these areas in the case group after 90 days compared to that in the control group (Figure 3, lower). The gross and microscopic examination of the liver tissue of all porcine models did not show any inflammation or abscess.

### 3.2. Body Weight and Serum Ghrelin Levels

The relative weight loss in the case group (*n* = 1) was compared to that in the control group (*n* = 3) following G-DFR placement (Figure 4). The differences between the control and case groups with respect to the relative weight loss were as follows: 3.2% (2.5 vs. −0.7) on day three, 5.0% (12.0 vs. 7.0) on day 14, 17.0% (27.3 vs. 10.3) on day 28, 26.8% (42.2 vs. 15.4) on day 42, 31.4% (57.7 vs. 26.3) on day 56, 56.9% (98.8 vs. 41.9) on day 70, and 55.4% (140.6 vs. 85.2) on day 90 after G-DFR placement (Figure 4).

In terms of relative weight loss between the control and case porcine groups without the migration of G-DFR during follow-up periods (observation periods: 28 days (*n* = 1), 42 days (*n* = 1), 56 days (*n* = 1), and 90 days (*n* = 1)), the results showed that the relative weight loss on days 70 (56.7%, *p* = 0.013) and 90 (55.4%, *p* = 0.017) was statistically significant in the case group (Appendix A).

With respect to ghrelin hormone level changes between control (*n* = 3) and case (*n* = 1) groups after G-DFR placement for 90 days, the level of ghrelin in the case group was relatively higher compared to that in the control group; A significant difference was observed on days 28 (*p* = 0.049) between control and case group (Figure 5). Regarding the differences in ghrelin hormone levels between the control and the case group observed in the mid-term follow-up group (observation period: 56 days (n = 1), 90 days (n = 1)), the ghrelin hormone level was significantly different between the control and case groups on days 3 (*p* = 0.030) and 28 (*p* = 0.012) after the placement of G-DFR (Appendix A).

### 3.3. Serum Glucose and Insulin Levels

Blood glucose levels were compared between the control group (*n* = 3) and case group (*n* = 1) following G-DFR placement. The blood glucose levels between the control (91 mg/dL) and case (42 mg/dL) groups showed significant difference on day 42 (*p* = 0.026) (Figure 6). This trend was subsequently maintained at 90 days.

In terms of changes in insulin hormone levels between the control group (*n* = 3) and case (*n* = 1) groups following G-DFR placement for 90 days, the case group showed a tendency of lower insulin levels compared to the control group without statistical significance (Figure 7).

### 3.4. Biochemical and Liver Fibrosis Analysis

Biochemical examination was performed to analyze cholesterol and ALT levels in this exploratory porcine (three animals in the control group and two in the case group) study. Upon comparison of ALT levels between the control group (*n* = 3) and case group (*n* = 1) following G-DFR placement for 90 days, compared to the control group, the case group showed a tendency of lower ALT levels without statistical significance. The level of ALT in the mid-term follow-up group (56-day and 90-day observation periods) was significantly lower than that in the control group (*p* = 0.005) (Appendix A, left). No significant difference was observed with respect to the cholesterol levels between the two groups (Appendix A, right).

Collagen formation was assessed as a marker of liver fibrosis via MT staining of the liver. The formation was significantly lower in the case group (*n* = 1) than in the control group following G-DFR placement for 90 days (1.40 vs. 2.92% as area of fraction object, *p* = 0.019) (Figure 8).

## 4. Discussion

In the current study, we evaluated the feasibility, safety, and efficacy of an experimentally woven G-DFR, a self-expandable gastro-duodenal metal stent that was partially covered with PTFE and distally attached to a PTFE skirt with a length of 60 mm. Endoscopic placement of G-DFR at the pyloroduodenal junction was technically effortless in all eight pigs, including those in the migration (*n* = 3) and mid-term efficacy (*n* = 5) groups. G-DFRs in five out of eight (63%) pigs migrated during the follow-up period. Only one animal in the efficacy group did not show G-DFR migration until the end of the follow-up period of 90 days. In all three cases of G-DFR without migration, stents were easily removed at preplanned intervals (28, 42, and 90 days). No adverse events were observed, and necropsy showed no evidence of severe histological inflammation in all pigs. 

Because of the relatively high migration rate of G-DFR, we investigated the efficacy of G-DFR by comparing the outcomes of the mid-term efficacy group (two pig without stent migration in 56 days and 90 days) with those of control groups (*n* = 3). Relative weight loss at 90 days was as high as 55.4%. In addition, the mid-term efficacy group showed significantly lower level of serum glucose from six weeks after G-DFR implantation until the end of the follow-up period of 90 days. The mid-term efficacy group showed a tendency of lower levels of insulin than the control group; however, the difference was not statistically significant, reflecting improvement in insulin sensitivity. Furthermore, significant improvements in hepatic parameters, including ALT levels and liver fibrosis, were observed in a case with well positioned G-DFR in the pyloroduodenal portion; however, the number of animals demonstrating the aforementioned findings was small. 

The level of serum ghrelin was higher in the case group than in the control group. Ghrelin is an orexigenic hormone that is released primarily from neuroendocrine cells located in the gastric fundus and partially from other organs, including duodenum [14,15,16]. However, the role of ghrelin in weight reduction in patients undergoing bariatric surgery or EBMTs has not been completely elucidated. The level of ghrelin decreases after sleeve gastrectomy, which involves removal of the gastric fundus, the primary source of ghrelin [17]. However, studies examining intragastric balloon or Roux-en-Y gastric bypass showed variable results with respect to the changes in levels of ghrelin [18,19]. Increase in ghrelin levels post bariatric surgery or EMBT might be associated with negative caloric balance [20]. The mechanism of weight reduction mediated by the G-DFR system might involve complex processes of neuroendocrine signaling attributed to delayed gastric emptying and bypass of the proximal duodenum. Further studies are required to understand the association between ghrelin level and relative weight loss following G-DFR implantation.

The novel G-DFR showed the following theoretical advantages for human trial. First, the endoscopic placement of G-DFR was more similar to that of conventional enteric metal stent than the over the wire placement of other EMBT devices alongside endoscopic guidance resulting in procedural convenience. Second, a previous study showed better in vitro performance of G-DFR with 15 mm inner diameter and distal 60 mm PTFE skirt compared to that of the intragastric balloon in terms of delayed gastric emptying for EBMT. Third, this G-DFR with a flexible 60 mm distal PTFE skirt in the proximal duodenum may not compromise the major duodenal papilla in the second portion of the duodenum in humans compared with previous duodenojejunal bypass liner showing frequent liver abscess or other adverse events attributed to the long length of the bypass liner [21]. Fourth, endoscopic stent placement for three months and removal following new G-DFR placement in the same session may be an attractive alternative for durable outcomes of EBMT. This G-DFR comprised both uncovered flaps in the proximal and distal region for anchoring to the pyloric and duodenal mucosa. No severe mucosal or submucosal injury or perforation was observed. Moreover, the G-DFR can be removed in three months after the placement. Tissue hyperplasia in the uncovered portion can be detached and removed via the shrinkage of G-DFR wire by grasping and pulling the lasso in the proximal flaps. Recently, medical nutritional therapy or novel interventions altering the microbiome has been introduced as a promising treatment for obesity and diabetes [22,23,24,25]; therefore, a multidisciplinary approach with medical nutritional therapy or novel interventions and EBMT including G-DFR may be evaluated in clinical trials.

This study has several limitations. The major limitation is frequent migration of G-DFR. Nearly every animal experienced migration, which would require endoscopic or possible surgical removal in humans. Even though the partial migration could be endoscopically removed, and fully migrated G-DFR without becoming lodged in the bowel in this pilot animal study, therefore, the refinement of G-DFR without migration may be required in the further study. We did not evaluate stenosis/stricture of the pylorus and proximal duodenum because necropsy was immediately performed after endoscopic removal of G-DFR. In case of migrated G-DFR, however, no stenosis and stricture were seen during endoscopic evaluation (Appendix A). Further long-term study may be required for evaluating stenosis/stricture of the pylorus and proximal duodenum after endoscopic removal of G-DFR. In this study, only two pigs made it the full way without migration. It is difficult to draw any meaningful conclusions from results, even though this is a pilot study. The novel G-DFR was experimentally cross-wired, and inconsistent heat treatment of a nitinol wire was performed by relatively inexperienced personnel. Therefore, uneven length and diameter of meshed G-DFR was observed (Appendix A). PTFE coating with the attachment of distal PTFE skirt in the wire mesh of G-DFR and the loading of G-DFR to the stent introducer was then referred to the manufacturer of enteric stent in this study. In situ placement of G-DFR in the present study demonstrated gradual narrowing of the opening in the proximal portion due to low radial force attributed to cross-wired woven and non-homogeneous experimental heat treatment of nitinol meshed in G-DFR during the follow-up period. This may affect migration of G-DFR by reducing the anchoring force in the proximal flap of G-DFR. Given the relatively frequent migration of G-DFR but promising results with respect to relative weight loss and lowering of serum glucose levels as well-positioned G-DFR in pyloroduodenal portion, this pilot study on G-DFR may just pave the way for the novel endoscopic platform of EBMT using a concept similar to that of the endoscopic placement of gastroduodenal enteric stent. Further refinement of G-DFR with enhanced wire structure and heat treatment of the wire mesh for the prevention of migration by a specialized manufacturer of enteric stent is underway.

## 5. Conclusions

G-DFR developed in this study showed technical feasibility and safety in a porcine model. G-DFR might induce relative weight loss and mediate metabolic improvements by multiple mechanisms, including delayed gastric emptying and bypass of proximal duodenum. We believe that the refinement of the structure of G-DFR will contribute to lowering the risk of G-DFR migration. G-DFR may have the potential as a promising alternative option in the endoscopic treatment of obesity and metabolic disorders.

## 6. Patents

Do Hyun Park is a listed inventor on an issued patent for a gastroduodenal flow restrictor (G-DFR) owned by the University of Ulsan Foundation for Industry Cooperation and the Asan Foundation. 

## Figures and Tables

**Figure 1 nutrients-14-02563-f001:**
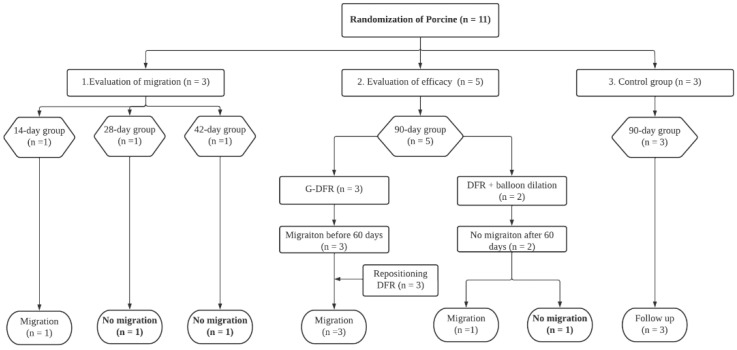
Flow diagram showing the steps of G-DFR placement. G-DFR, Gastro-duodenal flow restrictor.

**Figure 2 nutrients-14-02563-f002:**
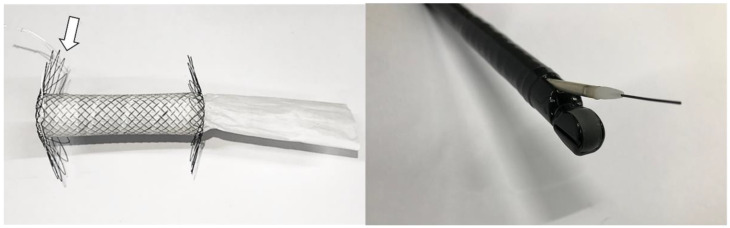
Photograph of G-DFR with proximal lasso (arrow) and 60 mm PTFE skirt. An over the wire of the stent introducer with 10.5 Fr outer diameter in an echoendoscope with 3.8 mm diameter of working channel. G-DFR, Gastro-duodenal flow restrictor; PTFE, polytetrafluoroethylene.

**Figure 3 nutrients-14-02563-f003:**
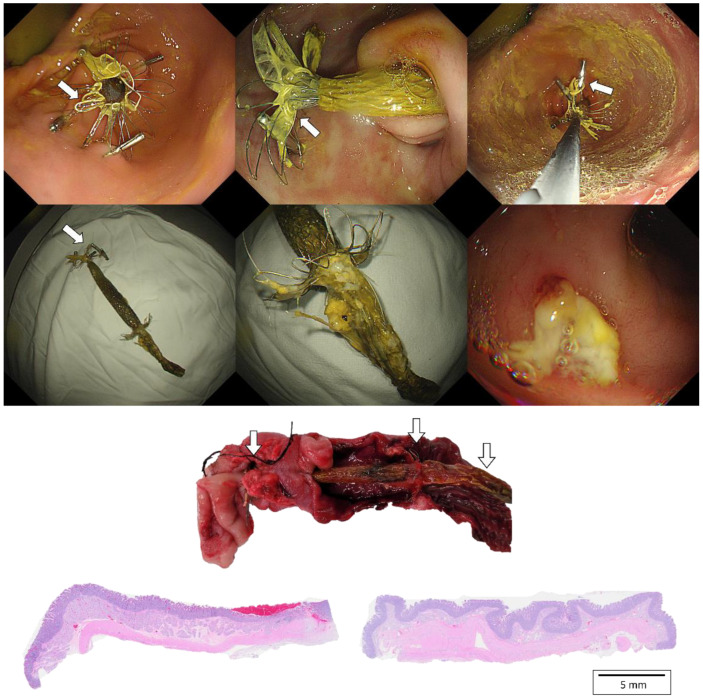
(**Upper**). Endoscopic images of a porcine stomach after deployment of the novel G-DFR. After 90 days of observation, G-DFR was then endoscopically removed. A white arrow indicates a lasso in proximal flaps. (**Middle**). Gross findings showing duodenum with retained G-DFR after necropsy. Both duodenum sections were selected approximately 5 cm away (white arrow) from the pylorus valve where the distal flap of G-DFR was loaded. (**Lower**). Representative histopathological images of the hematoxylin and eosin staining of the duodenum sections collected after 90 days from the case group (**left**) treated with G-DFR and the control group (**right**). In the case group, the duodenum showed mild ulceration and granulomatous formation but no perforation where the distal flap of G-DFR was loaded. G-DFR, Gastro-duodenal flow restrictor.

**Figure 4 nutrients-14-02563-f004:**
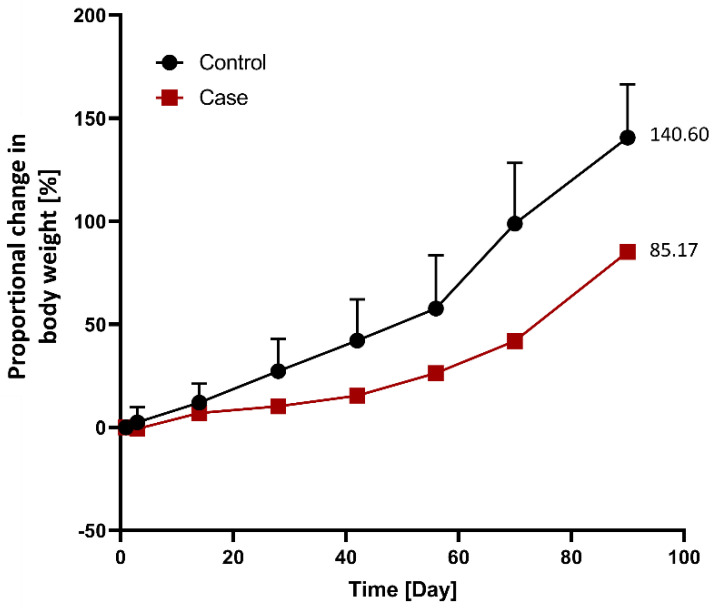
Comparison of the proportional changes in body weight between the control (*n* = 3) and case (*n* = 1) groups following G-DFR placement based on observation periods. The relative weight loss in the case group compared to that in the control group was as follows: 3.2% (2.5 vs. −0.7) at 3 days, 5.0% (12.0 vs. 7.0) at 14 days, 17.0% (27.3 vs. 10.3) at 28 days, 26.8% (42.2 vs. 15.4) at 42 days, 31.4% (57.7 vs. 26.3) at 56 days, 56.9% (98.8 vs. 41.9) at 70 days, and 55.4% (140.6 vs. 85.2) at 90 days after G-DFR placement. G-DFR, Gastro-duodenal flow restrictor.

**Figure 5 nutrients-14-02563-f005:**
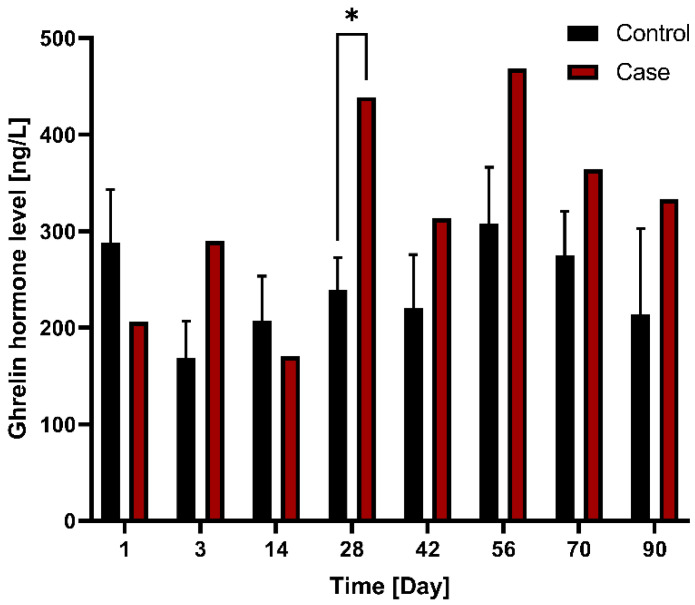
Comparison of changes in ghrelin hormone levels between the control (*n* = 3) and case (*n* = 1) groups with G-DFR placement for 90 days. A significant difference (*, *p* < 0.05) was observed in the ghrelin levels between control (239.5 ng/L) and case group (438.8 ng/L) on day 28 (*p* = 0.049). G-DFR, Gastro-duodenal flow restrictor.

**Figure 6 nutrients-14-02563-f006:**
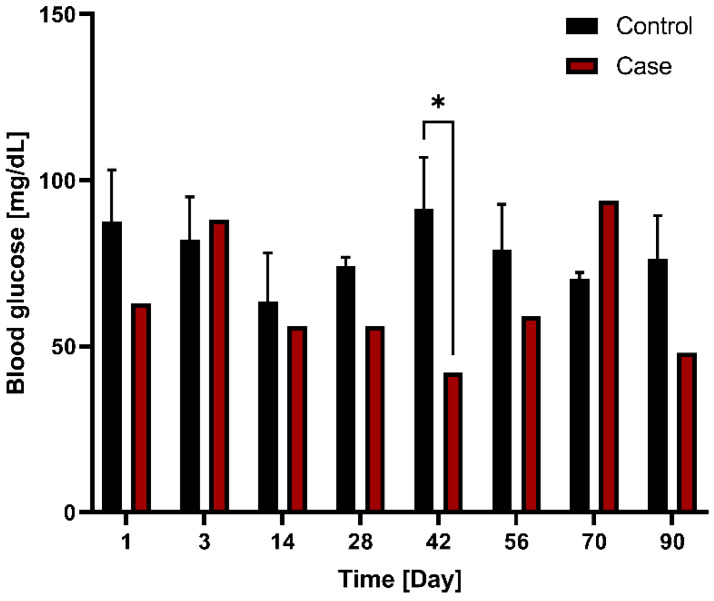
Comparison of blood glucose levels between the control (*n* = 3) and case (*n* = 1) groups with G-DFR placement based on observation periods. A significant difference (*, *p* < 0.05) was observed in the blood glucose levels between control (91 mg/dL) and case group (42 mg/dL) on day 42 (*p* = 0.026). This trend was subsequently maintained at 90 days. G-DFR, Gastro-duodenal flow restrictor.

**Figure 7 nutrients-14-02563-f007:**
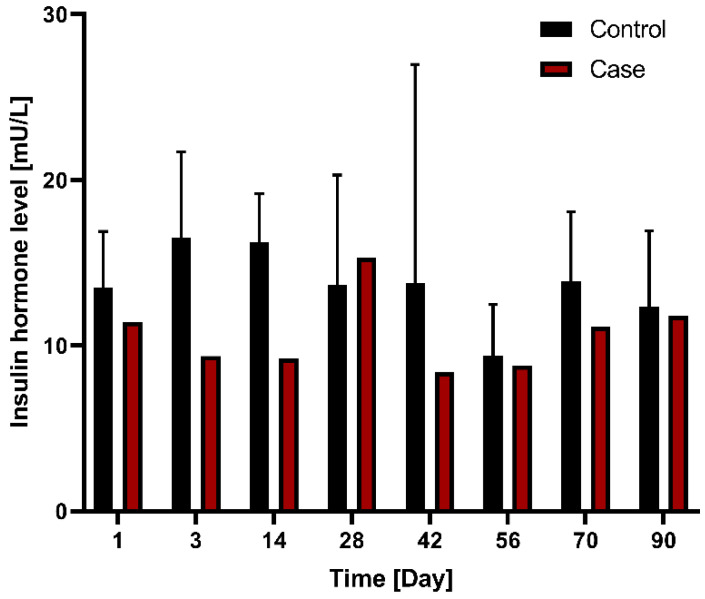
Comparison of changes in insulin hormone levels between the control group (*n* = 3) and case group (*n* = 1) with G-DFR placement for 90 days. G-DFR, Gastro-duodenal flow restrictor.

**Figure 8 nutrients-14-02563-f008:**
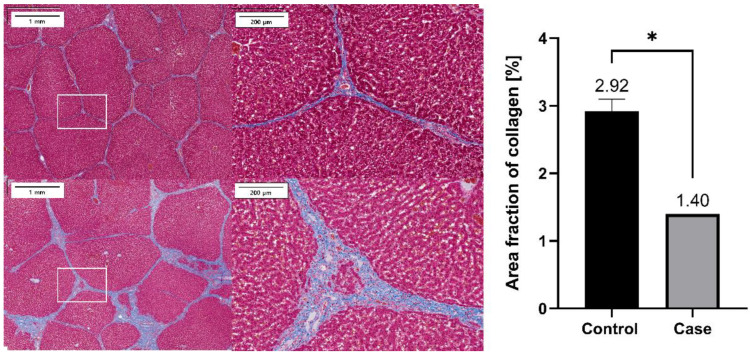
Representative histopathological images of the MT staining of the liver sections harvested from the case group (**upper**) and the control group (**below**). Percentage area of collagen (% of positively stained area) was compared between the control (*n* = 3) and case (*n* = 1) groups, which showed significant difference (*, *p* < 0.05) between the control and case groups (*p* = 0.019). MT: Masson’s Trichrome staining.

## Data Availability

Not applicable.

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
