# Peer review of "Effect of Novel Gastro-Duodenal Flow Restrictor on Relative Weight Loss and Glucose Levels in a Porcine Model: A Pilot Randomized Study"

_nutrients, 2022, doi:10.3390/nu14132563_

Round 1

Reviewer 1 Report

A few comments.

-How does the 15mm lumen diameter of the stent restrict flow? Is there further narrowing of the lumen within the stent? This needs to be explained further.

-How long were the pigs survived after placement of the stent? One concern following stent removal may be stenosis/stricture of the pylorus and proximal duodenum.

-Overall, the major limitation is migration. This is significant. Nearly every animal experienced migration which would require endoscopic or possible surgical removal in a human. This significant AE should not be understated.

-Any pigs with hepatic abscesses? 

-Im not certain how to interpret the weight loss in the context of device migration. Did the pigs lose weight because of the functionality of the device or because the device migrated and created essentially a small bowel obstruction? Difficult to interpret.

-Were the devices removed once migration was noted? If so, how would device pigs still be exhibiting reduced weight at day 56 and 70?

-Only two pigs made it the full way without migration. It is difficult to draw any meaningful conclusions on this study in that situation, even understanding this is a pilot study.

Author Response

Point 1: How does the 15 mm lumen diameter of the stent restrict flow? Is there further narrowing of the lumen within the stent?

Response 1: Thank you for your comment. A higher delayed gastric emptying time has been reported with a gastroduodenal stent of 18 mm diameter compared with a stent of 20 mm diameter (Endosc Int Open. 1(1):17-23 2013). Therefore, a gastroduodenal stent with 15 mm lumen diameter may further restrict flow of gastric contents and delay gastric emptying. We attached a 60 mm PTFE skirt in the distal part of the stent as a flow restrictor. This may further restrict gastro-duodenal flow due to friction. In a previous study on in vitro evaluation with a 3D human peristaltic gastroduodenal simulator, the gastric emptying volume of G-DFR with 15 mm lumen diameter and a 60 mm PTFE skirt at the distal end of G-DFR was the least, suggesting enhanced gastric retention compared with intragastric balloon, gastroduodenal stents of 18 and 20 mm lumen diameter, and G-DFR with a 30 mm PTFE skirt. Therefore, we chose the 15 mm lumen diameter of G-DFR with 60 mm PTFE skirt for further delaying gastric emptying in this pilot animal study. For clarity, we updated the Methods section as follows:

“A higher delayed gastric emptying time has been reported with a gastroduodenal stent of 18 mm diameter compared with a stent of 20 mm diameter [11]. Therefore, a gastroduodenal stent with 15 mm lumen diameter may further restrict flow of gastric contents and delay gastric emptying. We attached a 60 mm PTFE skirt in the distal part of the stent as a flow restrictor. This PTFE skirt may further restrict gastro-duodenal flow due to friction (Figure 2, left). In a previous study on in vitro evaluation with a 3D human peristaltic gastroduodenal simulator [12], the gastric emptying volume of G-DFR with 15 mm lumen diameter and a 60 mm PTFE skirt at the distal end of G-DFR was the least, suggesting enhanced gastric retention compared with intragastric balloon, duodenal stents of 18 and 20 mm lumen diameter, and G-DFR with a 30 mm PTFE skirt. Therefore, we chose the 15 mm lumen diameter of G-DFR with 60 mm PTFE skirt for further delaying gastric emptying in this pilot animal study.”

Point 2: How long were the pigs survived after placement of the stent?

Response 2: Thank you for your careful comment. The pigs were followed up based on 14 days (evaluation of migration n=1), 28 days (evaluation of migration n=1), 42 days (evaluation of migration n=1), and 90 days (evaluation of efficacy n=5) observation after placement of G-DFR. All pigs survived until end of follow-up in each group, except those with migration. We mentioned this in the Methods section.

Point 3: One concern following stent removal may be stenosis/stricture of the pylorus and proximal duodenum.

Response 3: Thank you for excellent your comment. We did not evaluate stenosis/stricture of the pylorus and proximal duodenum because necropsy was immediately performed after endoscopic removal of G-DFR. In case of migrated G-DFR, however, no stenosis and stricture were seen during endoscopic evaluation. Further long-term study may be required for evaluating stenosis/stricture of the pylorus and proximal duodenum after endoscopic removal of G-DFR. We added these sentences in the Discussion section.

“We did not evaluate stenosis/stricture of the pylorus and proximal duodenum because necropsy was immediately performed after endoscopic removal of G-DFR. In case of migrated G-DFR, however, no stenosis and stricture were seen during endoscopic evaluation (Figure S4). Further long-term study may be required for evaluating stenosis/stricture of the pylorus and proximal duodenum after endoscopic removal of G-DFR.”

Supplementary Figure 4. Endoscopic images of porcine stomach (left) and proximal duodenum (right) while partial-migrated G-DFR was repositioned

Point 4: Overall, the major limitation is migration. This is significant. Nearly every animal experienced migration which would require endoscopic or possible surgical removal in a human. This significant AE should not be understated.

Response 4: We absolutely agree with your opinion that migration is major limitation in this pilot study. Nearly every animal experienced migration, which would require endoscopic or possible surgical removal in humans. Even though the partial migration could be endoscopically removed, fully migrated G-DFR was not lodged in the bowel in this pilot animal study; therefore, the refinement of G-DFR without migration may be required in the further study. We have added the following sentences to the limitation section accordingly:

“The major limitation is frequent migration of G-DFR. Nearly every animal experienced migration, which would require endoscopic or possible surgical removal in humans. Even though the partial migration could be endoscopically removed, and fully migrated G-DFR without becoming lodged in the bowel in this pilot animal study, therefore, the refinement of G-DFR without migration may be required in the further study.

Point 5: Any pigs with hepatic abscesses? 

Response 5: Thank you for your comment. All pigs maintained normal body temperature and good appetite during the observation periods. Moreover, the gross and microscopic examination of liver tissue of all porcine models did not show any inflammation or abscess. We have added this in the Results section as follows:

“The gross and microscopic examination of the liver tissue of all porcine models did not show any inflammation or abscess.”

Point 6: I’m not certain how to interpret the weight loss in the context of device migration. Did the pigs lose weight because of the functionality of the device or because the device migrated and created essentially a small bowel obstruction? Difficult to interpret.

Response 6: Thank you for your comment. Because of the difficult interpretation, cases involving migration were excluded, and cases without the migration of G-DFR (n=1 in 90 days as a primary outcome; n=1 in 90 days and n=1 in 56 days as a mid-term efficacy outcome shown in Supplementary results) were only evaluated for the interpretation of efficacy of G-DFR. In addition, no distally migrated device resulting in small bowel obstruction was observed during the follow-up periods. We have added these sentences in the Results section as follows:

“Because of difficult interpretation, case with migration was excluded and case without the migration of G-DFR (n=1 in 90 days as primary outcome, n=1 in 90 days and n=1 in 56 days as mid-term efficacy group with supplementary results) in was only evaluated for the interpretation of efficacy on G-DFR.”

Point 7: Were the devices removed once migration was noted? If so, how would device pigs still be exhibiting reduced weight at day 56 and 70?

Response 7: Thank you for your comment. Once the migration was noted during observation, partially migrated G-DFR in the proximal or distal duodenum was endoscopically removed using rat-tooth forceps. In addition, radiographic and biochemical examinations were conducted every two weeks until the end of the planned follow-up period. The interpreted data representing the ideal observed trend in weight and biochemical parameters were only selected based on the monitoring time point before migration. In Figure S1 and S2, the porcine model without migration for 90 days (n=1) is shown at every time points including 56 days, 70 days, and 90 days. One pig showed well-positioned G-DFR at 56 days and migration at 70 days. Data after 56 days in this pig were not evaluated. This case was added to the mid-term efficacy group (Supplementary results). We added this part in the Methods section as follows:

After G-DFR placement, the pigs were fed after recovery from the anesthesia. The case pigs were followed up based on 14 days (evaluation of migration n=1), 28 days (evaluation of migration n=1), 42 days (evaluation of migration n=1), and 90 days (evaluation of efficacy n=5) observation after placement of G-DFR. All pigs survived until fulfillment of follow-up periods in each group except the case with migration. (Figure.1) The interpreted data representing the case trend of weight and biochemical value was only selected based on the monitoring time point before possible migration was noted.”

“Weight and behavioral changes and the levels of fasting blood glucose, ghrelin, insulin, alanine aminotransferase (ALT), and cholesterol were monitored after two days of G-DFR placement every two weeks until end of planned follow-up.”

Point 8: Only two pigs made it the full way without migration. It is difficult to draw any meaningful conclusions on this study in that situation, even understanding this is a pilot study.

Response 8: Thank you for your comment. We totally agree with your comment that only a small sample population made it the full way without migration. However, the final goal of this pilot study was to assess technical feasibility and safety in a porcine model. To verify and refine the tendency of the physiological results from the case group, further refinement of G-DFR to prevent migration is necessary. We have added the following into the Discussion section:

“In this study, only two pigs made it the full way without migration. It is difficult to draw any meaningful conclusions from results, even though this is a pilot study.“

Reviewer 2 Report

The manuscript submitted to Nutrients for publication by Huh et al., titled: "Effect of novel gastro-duodenal flow restrictor on relative weight loss and glucose levels in a porcine model: a pilot randomized study", is an in vivo study evaluating the effects and efficacy towards weight loss of a new method of bariatric intervention. This is an interesting topic with potentially significant clinical implications. 

The reviewer would like to offer the following points for consideration:

1. How was the number of the experimental animals determined (power calculation etc)

2. A discussion regarding the relationship between the microbiome and insulin resistance would be useful especially given the points regarding obesity and glucose control. A paper that might be of interest and helpful towards that end is the following: Sikalidis, A.K.; Maykish, A. The Gut Microbiome and Type 2 Diabetes Mellitus: Discussing A Complex Relationship. Biomedicines 2020, 8, 8. https://doi.org/10.3390/biomedicines8010008.

3. Briefly discussing nutritional options and dietary implications post application of the method would be interesting in the discussion section. There are certain types of nutrients and foods such as berries that are recommended in cases like these from a medical nutrition therapy clinical standpoint. A paper that could contribute to such a discussion is the following:

Kristo, A.S.; Klimis-Zacas, D.; Sikalidis, A.K. Protective Role of Dietary Berries in Cancer. Antioxidants 2016, 5, 37. https://doi.org/10.3390/antiox5040037.

4. Proofreading is recommended.

Nice work overall.

Author Response

Point 1: How was the number of the experimental animals determined (power calculation etc)

Response 1: Thank you for your comment. Because this was a pilot randomized study, the numbers of animals in the experimental and control group were not calculated. We have already mentioned this in the Statistical Analysis section of the Methods section as follows:

“Since this was a pilot randomized study, the sample size was not calculated.”

Point 2: A discussion regarding the relationship between the microbiome and insulin resistance would be useful especially given the points regarding obesity and glucose control. A paper that might be of interest and helpful towards that end is the following: Sikalidis, A.K.; Maykish, A. The Gut Microbiome and Type 2 Diabetes Mellitus: Discussing A Complex Relationship. Biomedicines 2020, 8, 8. https://doi.org/10.3390/biomedicines8010008.

Response 2: Thank you for your thoughtful comment. We updated the discussion section as follows:

Recently, medical nutritional therapy or novel interventions altering the microbiome has been introduced as a promising treatment for obesity and diabetes [22-25]; therefore, a multidisciplinary approach with medical nutritional therapy or novel interventions and EBMT including G-DFR may be evaluated in clinical trials.”

Point 3: Briefly discussing nutritional options and dietary implications post application of the method would be interesting in the discussion section. There are certain types of nutrients and foods such as berries that are recommended in cases like these from a medical nutrition therapy clinical standpoint. A paper that could contribute to such a discussion is the following:  Kristo, A.S.; Klimis-Zacas, D.; Sikalidis, A.K. Protective Role of Dietary Berries in Cancer. Antioxidants 2016, 5, 37. https://doi.org/10.3390/antiox5040037.

Response 3: Thank you for your suggestion. We updated the discussion section as follows:

Recently, medical nutritional therapy or novel interventions altering the microbiome has been introduced as a promising treatment for obesity and diabetes [22-25]; therefore, a multidisciplinary approach with medical nutritional therapy or novel interventions and EBMT including G-DFR may be evaluated in clinical trials.”

Round 2

Reviewer 1 Report

No additional revisions requested.

Reviewer 2 Report

The authors have made reasonable efforts in addressing the reviewer’s comments. Proofreading is recommended.